# Knowledge, attitude and practices of community pharmacists regarding COVID-19: A paper-based survey in Vietnam

**Huong Thi Thanh Nguyen[1], Dai Xuan Dinh[1]\*, Van Minh Nguyen[2]**

**1** Department of Pharmaceutical Management and PharmacoEconomics, Hanoi University of Pharmacy, Hanoi City, Vietnam, **2** Center for Population Health Sciences, Hanoi University of Public Health, Hanoi City, Vietnam

\* dinhxuandai.224@gmail.com

## Abstract

### Objective

To survey the knowledge, attitude, and practices of Vietnamese pharmacists regarding the COVID-19 pandemic.

### Method

This cross-sectional, paper-based study was conducted from June to August 2020. A validated questionnaire (Cronbach's alpha = 0.84) was used to interview 1,023 pharmacists in nine provinces of Vietnam. Analysis of covariance was employed to identify factors associated with the knowledge of pharmacists. The best model was chosen by using the Bayesian Model Averaging method in R software version 4.0.4.

### Results

The mean knowledge score was 12.02 ± 1.64 (range: 6–15), which indicated that 93.4% of pharmacists had good knowledge of COVID-19. There was no difference in the average score between males and females (p > 0.05). The multivariate linear regression model revealed that the knowledge was significantly associated with pharmacists' age, education level, and residence (p < 0.001). About attitude and practices, pharmacists daily sought and updated information on the COVID-19 pandemic through mass media and the internet (social network and online newspapers). Nearly 48% of them conceded that they communicated with customers when at least one person did not wear a face mask at the time of the COVID-19 outbreak. At medicine outlets, many measures were applied to protect pharmacists and customers, such as equipping pharmacists with face masks and hand sanitizers (95.0%), using glass shields (83.0%), and maintaining at least one-meter distance between two people (85.2%).

**Data Availability Statement:** All relevant data are within the manuscript and its Supporting information files.

**Funding:** The author(s) received no specific funding for this work.

**Competing interests:** The authors have declared that no competing interests exist.

## Conclusion

The pharmacists' knowledge of COVID-19 transmission, symptoms, and prevention was good. Many useful measures against the spread of this perilous virus were applied in medicine outlets. However, pharmacists should restrict forgetting to wear face masks in communication with medicine purchasers. The government and health agencies should have practical remedies to reduce the significant differences in the COVID-19 knowledge of pharmacists among provinces and education-level groups.

## Introduction

COVID-19 is an infectious disease caused by a newly discovered coronavirus. In Wuhan City in China, the first cases of COVID-19 contamination were detected in December 2019 [1]. After that, this perilous virus was quickly spread globally and gave rise to an outbreak of an international pandemic. Up to 2021 Jul 02, globally, there were approximately 180.49 million COVID-19 confirmed cases and 3,916,771 deaths. In the previous week, there were approximately 2.60 million new cases, and 57,603 deaths reported [2]. In Vietnam, up to 2021 Jul 02, there were 17,902 COVID-19 confirmed cases, 84 deaths, and 7,247 recoveries [3]. This pandemic begot many negative influences on local and global economies. Recently, many stringent measures have been implemented in countries, such as social distancing, complete lockdowns with the aim of curbing the COVID-19 spread [4, 5]. At this time, at least 13 kinds of vaccines have been administered all over the world. In addition, 105 vaccines are in clinical development and 184 vaccine candidates are in pre-clinical development [6].

Medicine outlets are commonly the reliable and first place for citizens to take advice about health issues. Along with doctors and nurses, community pharmacists can play an important role in putting a curb on the spread of the COVID-19 pandemic [7, 8]. They can bear responsibilities for being the rapid point of care tests for COVID-19, taking notes about suspected cases, ensuring the medicine quality, mitigating drug shortages, supplying reliable information on COVID-19 to the community, ensuring education and home care for citizens, suspected patients, and family members while in self-isolation [9]. In China, when the outbreak had just commenced, pharmacists quickly compiled documents to introduce detailed information on therapeutic drugs for COVID-19. Hence, doctors could master the characteristics and instructions of these medications [10]. In the United States, the active engagement of pharmacists in COVID-19 testing can assist in addressing many gaps in testing availability and demonstrating the value that pharmacists can provide in addressing unmet health needs [11]. In Vietnam, roughly 82.000 community pharmacists are working in 61.000 medicine outlets. There are three main groups of Vietnamese pharmacists licensed to sell medicines: the Uni group (pharmacists who graduated universities of pharmacy with five years of training), the Col group (assistant pharmacists who graduated colleges of pharmacy with three years of training), and the Mid group (middle pharmacists who finished two-year-training courses). Normally, Vietnamese pharmacists with higher levels of education (such as Master, Ph.D.) do not directly sell medicines in medicine outlets.

The KAP is a representative survey conducted on a particular population to identify the knowledge (K), attitudes (A), and practices (P) of that population on a specific topic [12]. This standardized method has been used for conducting abundant studies all over the world. Regarding COVID-19, globally, many studies on the KAP of pharmacists and pharmacy students were carried out [13–30]. Up to 2021 Jul 02, there are only two published articles assessing COVID-19 related KAP of healthcare workers in Vietnam [13, 14]. There is no previous

report of a nationwide study assessing the KAP of pharmacists about COVID-19. This study was conducted to survey the KAP of 1,023 Vietnamese community pharmacists regarding COVID-19 in the year 2020.

## Materials and methods

### Study design

This cross-sectional, paper-based survey was conducted in Vietnam during the COVID-19 outbreak (from June 01 to August 31, 2020). This study was approved by the ethical committee of Hanoi University of Pharmacy, Hanoi, Vietnam (reference number 3-20/PCT-HĐĐĐ).

### The questionnaire

After reviewing numerous articles in PubMed and the websites of the World Health Organization (WHO) [31–34], the research team developed the first draft of the questionnaire. To guarantee the clarity and plainness of questions, three senior lecturers of Hanoi University of Pharmacy assisted the research team in reviewing the questionnaire. Furthermore, to ensure logic, suitability, and validity, this questionnaire was employed to conduct a pilot study with 40 pharmacists working in 20 medicine outlets in Hanoi in April 2020 before interviewing pharmacists all over Vietnam. The Cronbach's alpha was 0.84. The final questionnaire started with a short introduction, including the objectives, the procedures, the declaration of confidentiality and anonymity, and the volunteer nature of the participants. It was designed in Vietnamese and included the following main parts: pharmacists' profile (5 questions), knowledge (15 questions), and attitude/practices (33 questions) (S1 Questionnaire).

### Data collection

The Raosoft sample size calculator was used for computing this study's sample size. To achieve a margin of error of 5%, the confidence level of 99%, the response distribution of 50%, and the population size of 82,000 pharmacists, the sample size was 659. With the aim of increasing the generalizability and validity of this research, we strived to approach as many pharmacists as possible to collect data. A total of 1,200 pharmacists were approached. The response rate was 85.25%. 1,023 pharmacists voluntarily answered the questionnaire and verbal informed consent was obtained from all of them. The study sample can be representative of Vietnamese pharmacists. The general characteristics of the participants are shown in Table 1. Pharmacists were mainly women (78.2%). Most of them were aged from 20 to 39 years old (75.8%). Their time of working in medicine outlets was mostly less than 15 years. Nearly 66% of pharmacists were from the northern part of Vietnam.

By virtue of the complexity of the COVID-19 plague, a non-probability convenience sampling technique was used for recruiting participants. Participants' inclusion criteria were licensed pharmacists working in medicine outlets opened on the days of data collection, able to read and understand Vietnamese, and at least 18 years old. In each province, data collection forms were distributed to pharmacists with the aid of medical personnel of the Department of Health (data collectors). At medicine outlets, each pharmacist was given one data collection form. After the forms were filled in, data collectors would check them to guarantee that all questions were answered adequately and legibly. Then, all data collection forms were sent to the research team by post.

**Table 1. The general characteristics of the study sample (n = 1,023 pharmacists).**

| No. | Characteristics | | | Number | (%) |
|-----|-----------------|---|---|--------|-----|
| A1 | Gender | Male | | 223 | 21.8 |
| | | Female | | 800 | 78.2 |
| A2 | Age | 20–29 | | 320 | 31.3 |
| | | 30–39 | | 455 | 44.5 |
| | | 40–49 | | 156 | 15.2 |
| | | ≥ 50 | | 92 | 9.0 |
| A3 | Education level | University and higher (Uni) | | 364 | 35.6 |
| | | College (Col) | | 394 | 38.5 |
| | | Middle (Mid) | | 265 | 25.9 |
| A4 | Working time in medicine outlets (years) | < 5 | | 367 | 35.9 |
| | | 5–9 | | 309 | 30.2 |
| | | 10–14 | | 182 | 17.8 |
| | | 15–19 | | 68 | 6.6 |
| | | ≥ 20 | | 97 | 9.5 |
| A5 | Region (province) | Northern | Caobang | 33 | 3.2 |
| | | | Hanam | 95 | 9.3 |
| | | | Hanoi | 124 | 12.1 |
| | | | Namdinh | 52 | 5.1 |
| | | | Tuyenquang | 193 | 18.9 |
| | | | Vinhphuc | 177 | 17.3 |
| | | Central | Thanhhoa | 173 | 16.9 |
| | | | Nghean | 26 | 2.5 |
| | | Southern | Hochiminh | 150 | 14.7 |

## Data analysis

The data were entered and analyzed using Microsoft Excel 2020 and R software version 4.0.4. The descriptive analysis was undertaken using mean and percentage. Data normality was checked using histogram, Q-Q plot, boxplot, and *Shapiro-Wilk test* (p-value > 0.05 indicating a normally distributed continuous variable). Differences among groups were analyzed using the *Wilcoxon rank-sum test*, the *Kruskal-Wallis rank-sum test*, and the *Dunn test* for multiple comparisons.

Analysis of covariance (ANCOVA), a general multivariate linear model, was employed to identify factors associated with pharmacists' knowledge scores. The model was chosen by using the Bayesian Model Averaging method. The score for each pharmacist was computed based on three main topics: the ways of viral transmission (three questions), related symptoms (eight questions), and COVID-19 prevention (four questions). The right and wrong answer for each question was assigned 1 point and 0 point, respectively. The total knowledge score of one pharmacist varied from 0 to 15. The score ≥ 10 and < 10 indicated good knowledge and poor knowledge, respectively. Correct answers were mainly compared to information from the WHO [31–34].

## Results

### The knowledge

Almost all pharmacists (> 93%) knew that the COVID-19 virus could be directly transmitted by close contact with infected people via secretions of the mouth and nose, and direct contact

**Table 2. The knowledge of Vietnamese pharmacists about COVID-19.**

| No. | COVID-19 knowledge | Right answers (%) |
|---|---|---|
| | K1. Ways of COVID-19 transmission (B1-B3): | |
| B1 | The virus that causes COVID-19 spreads primarily through droplets generated when an infected person coughs, sneezes, or speaks. | 1,003 (98.0) |
| B2 | People can become infected by touching a contaminated surface and then touching their eyes, nose, or mouth before washing their hands. | 886 (86.6) |
| B3 | Direct contact with blood is not one way of COVID-19 transmission. | 957 (93.5) |
| | K2. COVID-19 symptoms include (B4-B11): | |
| B4 | Fever (common symptom) | 1,016 (99.3) |
| B5 | Dry cough (common symptom) | 1,009 (98.6) |
| B6 | Tiredness (common symptom) | 867 (84.8) |
| B7 | Difficulty breathing and shortness of breath | 975 (95.3) |
| B8 | Sore throat | 461 (45.1) |
| B9 | Nausea and vomiting | 315 (30.8%) |
| B10 | Headache | 764 (74.7%) |
| B11 | Bellyache and diarrhea | 113 (11.0%) |
| | K3. How to prevent COVID-19 (B12-B15): | |
| B12 | Maintain at least one-meter distance with people coughing/sneezing | 993 (97.1) |
| B13 | Use a mask, cover your mouth and nose when coughing/sneezing | 1,005 (98.2) |
| B14 | Wash hands regularly with soap or alcohol-based hand rub, and not touch the face | 976 (95.4) |
| B15 | Supplement nutrition and regularly do exercise | 955 (93.4) |

with blood is not one way of viral transmission. About 13.4% of pharmacists did not know the indirect COVID-19 transmission (touching a contaminated surface and then touching eyes and nose). Some COVID-19 symptoms which numerous pharmacists ($> 84\%$) knew were fever, dry cough, tiredness, and difficult respiration. The percentages of pharmacists who knew less common symptoms of COVID-19 (nausea, vomiting, bellyache, and diarrhea) were low ($< 50\%$). In addition, most pharmacists ($> 93\%$) comprehended the main ways of COVID-19 prevention (Table 2).

The mean knowledge score of 1,023 pharmacists was 12.02 ± 1.64, which indicated that 93.4% of pharmacists had good knowledge about COVID-19. There was no difference in the average scores between males and females ($p > 0.05$, *Wilcoxon rank-sum test*). The average knowledge score of Col pharmacists was significantly higher than that of Uni and Mid pharmacists ($p < 0.05$, *Dunn test*). The average score of pharmacists in the central part of Vietnam was significantly higher than that in other parts ($p < 0.001$, *Dunn test*) (Table 3).

Among provinces, the average knowledge scores of pharmacists were seemingly equal, except for Thanhhoa and Tuyenquang province. The average score of pharmacists in Hanam, Hanoi capital, Hochiminh city, Thanhhoa, Tuyenquang, Vinhphuc, and other provinces was 11.60, 12.13, 12.34, 13.38, 11.02, 11.97, and 11.51, respectively. The knowledge score of pharmacists in Tuyenquang province was significantly lower than that in Hanam, Hanoi, Hochiminh, Thanhhoa, and Vinhphuc ($p < 0.001$, *Dunn test*). The knowledge score of pharmacists in Thanhhoa province was significantly higher than that in other provinces ($p < 0.001$, *Dunn test*) (Fig 1).

In the multivariate linear regression analysis, after controlling for other variables, age, education, and residence (provinces) were three factors associated with the knowledge scores of Vietnamese pharmacists ($p < 0.001$). Gender was not associated with the knowledge scores of

**Table 3. The average score involving COVID-19 knowledge of pharmacists.**

| No. | Characteristics | | Number | Average score (SD) | p-value |
|---|---|---|---|---|---|
| 1 | Gender | Male | 223 | 12.03 (1.39) | 0.660 |
| | | Female | 800 | 12.02 (1.70) | |
| 2 | Age | 20–29 | 320 | 12.02 (1.63) | 0.159 |
| | | 30–39 | 455 | 11.93 (1.67) | |
| | | 40–49 | 156 | 12.29 (1.53) | |
| | | ≥ 50 | 92 | 11.97 (1.65) | |
| 3 | Education level | University and higher (Uni) | 364 | 12.23 (1.33) | < 0.001 |
| | | College (Col) | 394 | 12.45 (1.60) | |
| | | Middle (Mid) | 265 | 11.09 (1.72) | |
| 4 | Working experience in medicine outlets (years) | < 5 | 367 | 11.92 (1.53) | < 0.001 |
| | | 5–9 | 309 | 12.06 (1.73) | |
| | | 10–14 | 182 | 12.36 (1.64) | |
| | | 15–19 | 68 | 11.87 (1.65) | |
| | | ≥ 20 | 97 | 11.76 (1.64) | |
| 5 | Region | Northern (Caobang, Hanam, Hanoi, Namdinh, Vinhphuc, Tuyenquang) | 674 | 11.60 (1.65) | < 0.001 |
| | | Central (Thanhhoa, Nghean) | 199 | 13.20 (1.29) | |
| | | Southern (Hochiminh) | 150 | 12.34 (1.06) | |

pharmacists. There is a strong relationship between the age and the working experience of pharmacists in medicine outlets (p < 0.001, *Spearman's rank correlation*) (Table 4).

## The attitude and practices

In general, almost all pharmacists (99.5%) quotidianly sought and updated COVID-19 information. Common sources of information included mass media (television, radio) and the

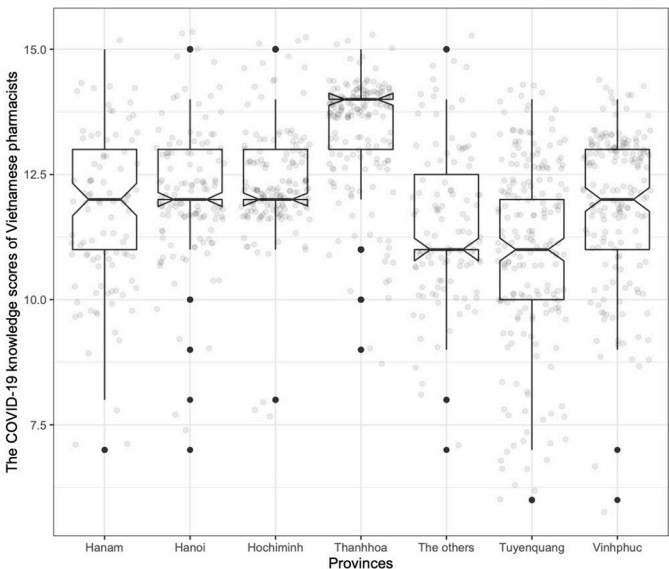

**Fig 1. The COVID-19 knowledge scores of Vietnamese pharmacists in provinces.**

**Table 4. Analysis of factors associated with the knowledge scores of Vietnamese pharmacists.**

| No. | Variables | Univariate linear regression | | | Multivariate linear regression | | | |
|---|---|---|---|---|---|---|---|---|
| | | Coef. | t-value | p-value | Coef. | Adjusted coef. | t-value | p-value |
| 1 | **Age** (continuous variable) | 0.002 | 0.512 | 0.608 | 0.018 | 0.104 | 3.506 | < 0.001 |
| 2 | **Gender** (ref.: Female) | | | | | | | |
| | Male | 0.011 | 0.086 | 0.932 | | | | |
| 3 | **Working experience** (continuous variable) | -0.005 | -0.63 | 0.529 | | | | |
| 4 | **Education level** (ref.: College) | | | | | | | |
| | Middle | -1.356 | -11.078 | < 0.001 | -0.941 | -0.252 | -7.839 | < 0.001 |
| | University and higher | -0.216 | -1.928 | 0.054 | 0.020 | 0.006 | 0.172 | 0.863 |
| 5 | **Residence (provinces)** (Ref: Caobang) | | | | | | | |
| | Hanam | 0.206 | 0.700 | 0.484 | 0.429 | 0.076 | 1.509 | 0.132 |
| | Hanoi | 0.735 | 2.575 | 0.010 | 0.866 | 0.172 | 3.141 | 0.002 |
| | Hochiminh | 0.946 | 3.375 | < 0.001 | 1.052 | 0.228 | 3.840 | < 0.001 |
| | Namdinh | -0.048 | -0.147 | 0.883 | 0.514 | 0.069 | 1.620 | 0.106 |
| | Nghean | 0.606 | 1.586 | 0.113 | 1.008 | 0.096 | 2.721 | 0.007 |
| | Thanhhoa | 1.987 | 7.178 | < 0.001 | 2.035 | 0.466 | 7.583 | < 0.001 |
| | Tuyenquang | -0.373 | -1.359 | 0.174 | -0.090 | -0.022 | -0.339 | 0.734 |
| | Vinhphuc | 0.572 | 2.070 | 0.039 | 0.526 | 0.122 | 1.976 | 0.048 |

Adjusted R-squared (for the multivariate linear regression): 0.2685.

PostProbs (Bayesian Model Averaging): 0.267.

coef.: coefficient, ref.: reference.

internet (social networks and online newspapers). Approximately 89.4% of pharmacists were questioned by customers about COVID-19. Popular questions included the ways of COVID-19 transmission, related symptoms, and prevention. If customers had symptoms of viral contamination, 91.0% of pharmacists would ask them about some information (such as symptoms, travel history) to take notes and then report the information to health agencies when necessary. Nearly half of pharmacists communicated with customers when at least one person did not wear a face mask. The most popular reason was that pharmacists forgot to wear face masks (79.2%). Recently, several kinds of pharmaceutical products were usually purchased by customers, including vitamins, dietary supplements, antiseptics, and hand sanitizers. Moreover, in medicine outlets, there were many measures applied to curb viral transmission, such as equipping pharmacists with face masks and hand sanitizers (95.0%), maintaining the minimum distance between pharmacists and customers (85.2%), and using glass shields (83.0%) (Table 5).

## Discussion

Results from previous studies show that pharmacists were and are making vital contributions to impeding the expansion of COVID-19 pandemics [35–38]. As a result, enhancing knowledge of COVID-19 for pharmacists is of paramount importance. The average knowledge score of Vietnamese pharmacists was 12.02 (out of 15). Our results demonstrated that a majority of Vietnamese pharmacists have good knowledge about COVID-19 (93.4%), which was consistent with the findings of studies conducted in Lebanon (> 90%) [15], India (85.3%) [16], Pakistan (84%) [17], Cairo (83%) [18] but far higher than the results of Addis Ababa (53.2%) [19] and Gondar, Ethiopia (63%) [20].

**Table 5. The attitude and practices of pharmacists about COVID-19.**

| No. | Questions | "Yes" answers (%) |
|---|---|---|
| C1 | Do you seek and update information on COVID-19 every day? | 1,018 (99.5) |
|  | Your sources of COVID-19 information (C2-C4): |  |
| C2 | Shares from other colleagues/pharmacists | 533 (52.1) |
| C3 | The internet: social network (Facebook, Messenger, Zalo apps), and online newspapers (websites of the government and the health agencies) | 882 (86.2) |
| C4 | Mass media (national news, radio, television) | 987 (96.5) |
| C5 | In the last three months, whether or not customers have asked you questions involving COVID-19 | 915 (89.4) |
|  | Common questions of customers (C6-C9): |  |
| C6 | The ways of viral transmission | 788 (77.0) |
| C7 | Related symptoms | 830 (81.1) |
| C8 | COVID-19 prevention | 857 (83.8) |
| C9 | What they should do if suspecting of being infected with the COVID-19 virus | 579 (56.6) |
|  | In the last three months, pharmaceutical products usually purchased in your medicine outlet include (C10-C13): |  |
| C10 | Vitamins and dietary supplements* | 978 (95.6) |
| C11 | Medicines for treating influenza and headache | 420 (41.1) |
| C12 | Painkillers and antipyretics | 364 (35.6) |
| C13 | Antiseptics and hand sanitizers | 842 (82.3) |
| C14 | Do you think taking notes about the information of people who purchase medicines treating cough, fever, and flu is necessary? | 946 (92.5) |
| C15 | Will you ask customers some questions if they have symptoms the same as COVID-19 viral contamination? | 931 (91.0) |
|  | You will ask them about (C16-C19): |  |
| C16 | Their symptoms | 889 (86.9) |
| C17 | Travel history | 738 (72.1) |
| C18 | People who they had close contact with | 731 (71.5) |
| C19 | Whether or not they update information on COVID-19 regularly | 752 (73.5) |
| C20 | Have you ever communicated with customers when at least one person did not wear face masks in the last three months? | 490 (47.9) |
|  | If you already did it, list your reason(s) (C21-C23): |  |
| C21 | Because of lacking face masks in my medicine outlet | 148 (14.5) |
| C22 | I forgot to wear face masks | 388 (37.9) |
| C23 | Customers did not wear face masks | 19 (1.9) |
| C24 | Is there any measure to limit viral contamination and protect pharmacists and customers in your outlet? | 996 (97.4) |
|  | Measures for COVID-19 prevention in your outlet include (C25-C33): |  |
| C25 | Install glass shields (barriers or partitions) | 849 (83.0) |
| C26 | Put hand sanitizers at the doors of the outlet and require customers to use them before entering the outlet | 969 (94.7) |
| C27 | Maintain at least one-meter distance between pharmacists and customers | 872 (85.2) |
| C28 | Maintain at least one-meter distance between two pharmacists | 722 (70.6) |
| C29 | Equip pharmacists face masks and hand sanitizers | 972 (95.0) |
| C30 | Reduce the number of pharmacists in the outlet | 664 (64.9) |
| C31 | Organize courses of training and sharing information on COVID-19 among pharmacists | 550 (53.8) |
| C32 | Indirectly give customers advice on medicines (through phones, messenger, social media) | 464 (45.4) |

(*Continued*)

**Table 5.** (Continued)

| No. | Questions | "Yes" answers (%) |
|---|---|---|
| C33 | Deliver medicines to customers' doors | 430 (42.0) |

* dietary supplements: products made in the form of capsules, pellets, tablets, glues, granules, powder, liquid, and other processed forms containing one or a combination of the following substances: Vitamins, minerals, amino acids, fatty acids, enzymes, probiotics, and other biologically active substances; active ingredients naturally derived from animals, minerals, and plants through extraction, isolation, concentration, and metabolism processes.

In Vietnam, the fact that the number of females is far higher than that of males is one typical characteristic of the pharmaceutical industry. The proportion of female pharmacists in our study was high (78.2%), in line with results from the studies of Jordan (78%) [21], Goa (79.5%) [22], Cairo (70%) [18], and Lebanon (85.2%) [15]. Although the number of females was far higher than that of males, there was no difference in the COVID-19 knowledge score between these two groups in Vietnam. This finding was similar to results from the United Arab Emirates [23], Saudi Arabia [24], Nepal [25], and Jordan [26]. In Saudi Arabia, Pakistan, Nepal, and the United Arab Emirates, age was not the factor associated with the knowledge score of pharmacists [23–25, 27]. In Vietnam, it seems that better knowledge scores were associated with higher age of participants ($\beta$ = 0.104, p < 0.001), compatible with the findings in Jordan [28]. In the light of the strong correlation between age and working experience, the latter was excluded from the multivariate linear regression model. This factor was still significantly associated with Vietnamese pharmacists' knowledge. In Gondar, Ethiopia, pharmacists with more than six years of experience had better knowledge about COVID-19 in comparison with their counterparts with less than six years of experience [20]. In the United Arab Emirates, the knowledge score of pharmacists with more than five years of experience was significantly higher than that of pharmacists with less than two years of experience [23].

Education level was not a factor significantly associated with the knowledge score of pharmacists in several countries like the United Arab Emirates [23], Punjab and Khyber Pakhtunkhwa of Pakistan [27], and Saudi Arabia [24]. However, in Vietnam, there were significant differences in the mean knowledge scores among three education-level groups. A special point is that the average score of graduates from pharmacy colleges (Col pharmacists) was significantly higher than that of graduates from pharmacy universities (Uni pharmacists). Their time of studying and training was three and five years, respectively. The important rationale behind this result is that in Vietnam, the curriculum of pharmacy colleges usually focuses on practical subjects involving opening medicine outlets, selling medications, addressing health issues in the community. The curriculum of pharmacy universities commonly focuses on theoretical and academic subjects, such as drug manufacturing, pharmacological, and clinical pharmacy.

In Vietnam, the knowledge regarding COVID-19 of pharmacists was significantly different by region and province (residence). The average score of pharmacists in Thanhhoa province is the highest (13.38) while that in Tuyenquang province is the lowest (11.02). Thanhhoa is a border province located in the central part of Vietnam, having a flourishing economy and healthcare system. This province is located on the main road connecting the north and the south of Vietnam. The government and health authorities attached special importance to setting up activities involving COVID-19 prevention and pandemic control in border provinces. Besides, the Thanhhoa Department of Health usually organizes training courses for pharmacists to broaden their knowledge. For Tuyenquang, this is a highland province with a low population

density. The economy, healthcare, and traffic systems in many villages of this province are still undeveloped. In addition, at the time of data collection, there were COVID-19 confirmed cases in Thanhhoa while there were no cases found in Tuyenquang. The lack of hands-on experience on COVID-19 prevention may be a reason for the low knowledge scores of pharmacists in Tuyenquang.

In Ethiopia, 71.1% of community pharmacists checked new updates on COVID-19 more than once per day [20]. In Lebanon, many pharmacists spent about one to two hours per day getting information about this pandemic [15]. In Vietnam, COVID-19 drew pharmacists' special attention when 98.8% of pharmacists said that they sought and updated information on COVID-19 on a daily basis. They used the internet (online apps and newspapers) and mass media as the main sources to seek COVID-19 information. In Jordan, the sources of information for pharmacists mostly came from general media, WHO reports, published research papers, and social networks [28, 29]. Mass media (television, radio), the internet, and social media (Facebook, WhatsApp) were also the main sources of information of pharmacists in India [16], Turkey [30], Ethiopia [19, 20], and Cairo [18]. In Vietnam, due to the incompetence in reading and understanding documents written in English (language barriers), most pharmacists accessed sources of COVID-19 information spoken or written in Vietnamese. In many provinces, pharmacists were required to participate in groups on apps, such as Zalo or Messenger. These online groups are the sites where pharmacists can share their news and experience, and health agencies can directly and quickly send accurate information on COVID-19 to all pharmacists. In addition, to disseminate correct information about COVID-19, many text messages were sent to all citizens by the government and the Ministry of Health. Information on COVID-19 from the WHO was also translated and propagandized through national news on television. Recently, customers have usually purchased vitamins, dietary supplements, antiseptics, and hand sanitizers. This result showed that the knowledge involving COVID-19 prevention of the public was fairly good and activities involving propagandizing COVID-19 prevention of the government may be effective.

Community pharmacists can take an important part in disseminating COVID-19 knowledge to the public. Roughly 89% of Vietnamese pharmacists told that customers asked them about topics involving COVID-19 (such as the ways of viral transmission, symptoms, and preventative measures). In addition, 92.5% of pharmacists thought that it is necessary to collect the information of customers purchasing medicines treating cough, fever, and flu. This activity can help health agencies to contact people with suspected COVID-19. At present, the Hanoi Department of Health required all pharmacists to collect information about customers purchasing the aforementioned medicines to trace and contact them if necessary. These beneficial activities played a considerable part in bringing about impressive achievements in the battle against the COVID-19 pandemic in Vietnam.

During the period of the COVID-19 outbreak, pharmacists were equipped with face masks and hand sanitizers. At many medicine outlets, glass shields were installed to restrict the spread of droplets from saliva/nose when pharmacists directly communicated with customers. Hand sanitizers were also put at the doors of medicine outlets, and purchasers were encouraged to use them before entering the outlets. These useful activities contributed to protecting both pharmacists and customers. However, 47.9% of pharmacists conceded that they communicated with customers when at least one person was not wearing face masks in the last three months. The main reason is that pharmacists forgot to use face masks. In Turkey, 72.6% of pharmacists said that they did not wear any kind of mask [30]. In Lebanon, 30% of pharmacists rarely worn masks [15]. There is no denying that continually wearing face masks for many hours can render anyone uncomfortable. When medicine outlets were devoid of customers, pharmacists usually took off their face masks. In case of not wearing face masks, Vietnamese

pharmacists could be still protected since 86.3% of medicine outlets had fixed glass shields which can assist in reducing the capacity of viral contamination.

To prevent the transmission of the COVID-19 virus, Vietnam had many useful and practical remedies, such as isolating infected individuals, tracing and quarantining their contacts. From April to June 2020, schools were closed and activities for large crowds (such as festivals, conferences) were canceled. Citizens were encouraged to stay at home to minimize exposure and viral transmission. More importantly, face masks and hand sanitizers have been highly encouraged to be used. To hinder the capacity of COVID-19 spread from other countries, several rigorous measures were imposed, including a temporary suspension of entry of all foreigners who have come from or transited through COVID-19 affected areas, and a new mandatory regulation that all incoming travelers to Vietnam have to be quarantined at centralized facilities for 14 days [39, 40]. With good knowledge about COVID-19, Vietnamese pharmacists could bear numerous responsibilities and make a considerable contribution in the battle against the COVID-19 pandemic.

## Limitations

This national survey is the first paper-based study conducted to investigate the KAP of pharmacists about COVID-19 in Vietnam with a large sample size. Many effective measures against the spread of this perilous virus mentioned in this article can be applied in other countries. Most of the recent KAP studies involving COVID-19 were online surveys. A paper-based study has several following strengths: generate higher response rates than an online survey, easily and suitably approach various kinds of respondents who are not technology savvy. However, this research has some limitations. Besides the lack of human resources, there were difficulties in data collection. Social distancing was employed two times in Vietnam in April and July of 2020 due to the discovery of many COVID-19 patients in metropolises. Hence, the time for data collection was quite long and the number of surveyed pharmacists in some provinces was low. Furthermore, questions about COVID-19 knowledge only focused on three topics: ways of COVID-19 transmission, symptoms, and prevention. The results may not reflect the comprehensive knowledge of pharmacists about this pandemic.

## Conclusions

The knowledge, attitude, and practices of Vietnamese pharmacists about COVID-19 were good. There are many practical measures to curb viral contamination at medicine outlets and protect both pharmacists and medicine purchasers. However, pharmacists should restrict forgetting to wear face masks in communication with medicine purchasers. Besides, health agencies should have solutions to enhance knowledge for pharmacists, thereby guaranteeing that among provinces and education-level groups, there is no significant difference in the COVID-19 knowledge of pharmacists.

## Supporting information

**S1 Questionnaire.**
(DOCX)

**S1 Data.**
(XLSX)

## Author Contributions

**Conceptualization:** Huong Thi Thanh Nguyen, Van Minh Nguyen.

**Data curation:** Dai Xuan Dinh.

**Formal analysis:** Dai Xuan Dinh.

**Investigation:** Huong Thi Thanh Nguyen, Dai Xuan Dinh, Van Minh Nguyen.

**Methodology:** Huong Thi Thanh Nguyen, Dai Xuan Dinh, Van Minh Nguyen.

**Project administration:** Van Minh Nguyen.

**Software:** Dai Xuan Dinh.

**Supervision:** Huong Thi Thanh Nguyen.

**Validation:** Huong Thi Thanh Nguyen.

**Visualization:** Dai Xuan Dinh.

**Writing – original draft:** Dai Xuan Dinh.

**Writing – review & editing:** Huong Thi Thanh Nguyen, Dai Xuan Dinh, Van Minh Nguyen.

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
