## [Decision Letter · Decision Letter 0]

13 Apr 2021

PONE-D-21-08000

Knowledge, attitude and practices of community pharmacists regarding COVID-19: a paper-based survey in Vietnam

PLOS ONE

Dear Dr. Dinh,

Thank you for submitting your manuscript to PLOS ONE. After careful consideration, we feel that it has merit but does not fully meet PLOS ONE’s publication criteria as it currently stands. Therefore, we invite you to submit a revised version of the manuscript that addresses the points raised during the review process.

We look forward to receiving your revised manuscript.

Kind regards,

Bing Xue, Ph.D.

Academic Editor

PLOS ONE

Journal Requirements:

3. Please include additional information regarding the survey or questionnaire used in the study and ensure that you have provided sufficient details that others could replicate the analyses.

For instance, if you developed a questionnaire as part of this study and it is not under a copyright more restrictive than CC-BY, please include a copy, in both the original language and English, as Supporting Information.

Moreover, please include more details on how the questionnaire was pre-tested, and whether it was validated.

4. In your Methods section, please provide additional information about the participant recruitment method and the demographic details of your participants.

Please ensure you have provided sufficient details to replicate the analyses such as:

a) the recruitment date range (month and year),

b) a description of any inclusion/exclusion criteria that were applied to participant recruitment,

c) a table of relevant demographic details,

d) a statement as to whether your sample can be considered representative of a larger population,

e) a description of how participants were recruited, and

f) descriptions of where participants were recruited and where the research took place.

Reviewers' comments:

Reviewer's Responses to Questions

**Comments to the Author**

1. Is the manuscript technically sound, and do the data support the conclusions?

Reviewer #1: Partly

Reviewer #2: No

2. Has the statistical analysis been performed appropriately and rigorously? 

Reviewer #1: No

Reviewer #2: I Don't Know

3. Have the authors made all data underlying the findings in their manuscript fully available?

Reviewer #1: Yes

Reviewer #2: Yes

4. Is the manuscript presented in an intelligible fashion and written in standard English?

Reviewer #1: Yes

Reviewer #2: No

5. Review Comments to the Author

Reviewer #1: The manuscript is an excellent topic within this time we are living in. It discusses the KAP of COVID-19 in Vietnamese pharmacist. The sample size is quite big although the different backgrounds of provinces can be a source of misrepresentation of the sample.

I have added my notes to the attached PDF file, but I would like to highlight two points of mine:

1) would the authors consider performing a regression analysis of independent variables to check which of these factors is affecting the pharmacists' knowledge the most?

2) would the authors clarify the difference between different provinces and different health professions a bit further.

please check the attachment

Reviewer #2: This paper discusses Vietnamese pharmacists’ knowledge, attitude and practices regarding the COVID-19 pandemic through administering a national paper-based survey. There were 1,023 responses that were analyzed to look at the demographics, knowledge, attitudes, and practices with patients through the studies time frame (June to August 2020). The authors analyzed the responses based on descriptive statistics to gather the results. There were several typos and grammatical issues, the sentences’ structure was confusing at times making it hard to keep a logical flow of thought. There were major issues with the manuscript, for example:

1. There is no report on the response rate from all pharmacists who were given the survey

2. There was a majority of female respondents and no analysis was done to show if this was a confounder.

3. Authors attributed Vietnam’s unique low number of COVID cases/deaths to two reasons “copious doctors and nurses have strongly fighted the uphill battle in the frontline. In addition, drugstores are commonly a reliable and first place for citizens to take advice about ailments”. The reviewer is not sure how Vietnamese doctors and pharmacists are achieving significantly better outcomes than their global counterparts who can claim are doing the exact same measures.

4. What does the P values listed in Table 3 signify?

5. In looking into attitudes and practices of pharmacists, the survey included mainly non-reliable sources like other colleagues, internet and media, it didn’t include any scientific primary or secondary literature sources, WHO reports, even governmental health agencies. The authors called it “good knowledge” in line 181.

6. The authors did not elaborate in the results or discussion why there was a question in the survey about the pharmaceutical products usually purchased in drugstores.

7. Table 4, C14: the reviewer is not sure of the meaning (or legality) when the authors mention asking patients about “medical declaration” when they purchase cold and flu medicine.

8. Line 150, 151 mentions statistics about Iran with two different percentages.

9. Line 155: What do authors mean by “possessing many nimble and proficient authorities”?

10. Line 170-171: The authors did not offer adequate explanation for their results on why assistant pharmacists with 3 years of training scored higher in COVID-19 knowledge than “Uni” with 5 years training, mentioning that “it seems that they (Unis) don’t care much about issues in the community in comparison with Col pharmacists”!

11. The reviewer thinks the statement in line 220 on how paper surveys “approach various kinds of respondents who are not technology savvy (like old people)” is considered ageism and should be modified or removed.

6. PLOS authors have the option to publish the peer review history of their article (what does this mean?). If published, this will include your full peer review and any attached files.

Reviewer #1: No

Reviewer #2: No

---

## [Author Response · Author response to Decision Letter 0]

27 May 2021

Journal Requirements

We endeavored to revise mistakes in our manuscript. 

We strived to correct typos and grammar. 

In addition, Ms. Chi Nguyen Phuong, a Research Ph.D. Student from the University of Groningen, Netherlands, assisted us in language editing.

Moreover, please include more details on how the questionnaire was pre-tested, and whether it was validated.

The original questionnaire written in Vietnamese was added in the Supporting Information (S1 Questionnaire).

The pre-test and validation of our questionnaire were added on page 5 (lines 93 - 98).

4. In your Methods section, please provide additional information about the participant recruitment method and the demographic details of your participants.

Please ensure you have provided sufficient details to replicate the analyses such as:

a) the recruitment date range (month and year),

b) a description of any inclusion/exclusion criteria that were applied to participant recruitment,

c) a table of relevant demographic details,

d) a statement as to whether your sample can be considered representative of a larger population,

e) a description of how participants were recruited, and

f) descriptions of where participants were recruited and where the research took place.

Information on the aforementioned requirements can be found on pages 5 - 7 (lines 92 - 124).

Review Comments to the Author

Reviewer #1

The manuscript is an excellent topic within this time we are living in. It discusses the KAP of COVID-19 in Vietnamese pharmacist. I have added my notes to the attached PDF file, but I would like to highlight two points of mine:

1. Would the authors consider performing a regression analysis of independent variables to check which of these factors is affecting the pharmacists' knowledge the most?

We used the analysis of covariance (ANCOVA), a general multivariate model to identify factors associated with pharmacists’ knowledge scores.

The model reported in Table 4 was the best model chosen by using the Bayesian Model Averaging method (packages BMA and BAS in R software version 4.0.4). 

2. Would the authors clarify the difference between different provinces and different health professions a bit further.

We added some information and explanations for these results in the Discussion section (lines 222 - 253).

3. Could the difference in scores be due to years of experience? Regression analysis of different independent variables affecting knowledge score would be a good tool to find out.

When working experience, a continuous variable, is divided into five subgroups, there are differences in knowledge scores among subgroups: less than 5 years, 5-to-9, 10-to-14, 15-to-19, and ≥ 20 years (p < 0.001) (Table 3). However, in the multivariate linear model (Table 4), this factor was excluded because there is a strong relationship between the age and the working experience of pharmacists in medicine outlets (p < 0.001, Spearman’s rank correlation). We used the Bayesian Model Average method for selecting the best model (income variables include age, gender, education level, working experience, and residence). The results from this model show that the knowledge score was significantly associated with pharmacists’ age, education level, and residence (p < 0.001).

4. Line 124: Sometimes there are confusions about what is meant with "functional foods" and some might use the term "nutraceuticals”. Can the authors give a clear definition of what is meant with "functional foods" or to give examples in order to clarify. 

We used “Dietary Supplements” to supersede “Functional foods”. The definition of dietary supplements was added at the end of Table 5.

In 2014, the Vietnam Ministry of Health published a Circular regulating the management of functional foods. The definition of the dietary supplement was taken from this Circular. https://vanbanphapluat.co/circular-no-43-2014-tt-byt-regulating-the-management-of-functional-foods

5. This study included dentists. I believe that there are Jordanian studies about pharmacists. The other referenced studies within this paragraph includes nurses and healthcare workers and not pharmacists. The comparison would be better represented if done among studies including pharmacists.

References involving healthcare workers not pharmacists were removed. 

In addition, many previous studies on pharmacists were added [reference 15 - 30].

6. Results (line 99 - 104): Sentences can be clarified. Add numerical percentage for clarity.

Line 158 – 159, 179 - 184: Paraphrase the sentence to clarify. 

Line 186 – 187, 211 – 212: Could "not inconsiderable" be stated as considerable. The double negatives are confusing.

We revised these mistakes.

 

Reviewer #2: 

This paper discusses Vietnamese pharmacists’ knowledge, attitude and practices regarding the COVID-19 pandemic through administering a national paper-based survey. There were 1,023 responses that were analyzed to look at the demographics, knowledge, attitudes, and practices with patients through the studies time frame (June to August 2020). The authors analyzed the responses based on descriptive statistics to gather the results. There were major issues with the manuscript, for example:

1. There is no report on the response rate from all pharmacists who were given the survey.

This information can be found in lines 108 - 110. 

2. There was a majority of female respondents and no analysis was done to show if this was a confounder. 

What does the P values listed in Table 3 signify?

We used the analysis of covariance (ANCOVA), a general multivariate linear model, to identify factors associated with the knowledge scores of pharmacists. The model was chosen by using the Bayesian Model Averaging method. The old Table 3 was removed. New results can be seen in Table 3 and Table 4.

3. Authors attributed Vietnam’s unique low number of COVID cases/deaths to two reasons “copious doctors and nurses have strongly fighted the uphill battle in the frontline. In addition, drugstores are commonly a reliable and first place for citizens to take advice about ailments”. The reviewer is not sure how Vietnamese doctors and pharmacists are achieving significantly better outcomes than their global counterparts who can claim are doing the exact same measures.

We revised this mistake. We listed some possible reasons explaining why the COVID-19 confirmed cases and deaths in Vietnam were low. We did not mean to compare the roles and achievements among health workers or countries. We are truly sorry for this misunderstanding. 

4. In looking into attitudes and practices of pharmacists, the survey included mainly non-reliable sources like other colleagues, internet and media, it didn’t include any scientific primary or secondary literature sources, WHO reports, even governmental health agencies. The authors called it “good knowledge” in line 181.

We added some information on pages 14, 19, and 20. 

In the data collection form, for questions that we think that pharmacists can have other answers to (such as the question about sources of COVID-19 information), there are some blank rows in which pharmacists can write their additional answers.

In districts/provinces, community pharmacists are required to participate in online groups (on apps, such as Messenger and Zalo). These online groups are sites where pharmacists can share their news and experience, and health agencies can directly and quickly send accurate information on COVID-19 to all pharmacists. In addition, monthly, health agencies commonly organize meetings/courses to enhance the knowledge and practices of pharmacists. 

In Vietnam, in fact, due to the language barriers, a majority of pharmacists only access sources of COVID-19 information spoken or written in Vietnamese. WHO reports and scientific literature sources spoken or written in English will be translated and propagandized through national news on television (by the government and health agencies). In addition, Facebook is extremely popular in Vietnam. The government and health agencies have some official pages on Facebook (for example https://www.facebook.com/tintucvtv24/), and accurate and reliable information can be disseminated to not only pharmacists but also the public. For online newspapers, the government has the website “vnexpress.net”. Everyone can find news and related information on COVID-19 from these sources. Last but not least, information on COVID-19 was strictly controlled by the government. For example, if a person posts inaccurate or controversial information on Facebook, he/she will be quickly fined. 

Each person has their favorite sources to seek information they need. We think that people aged 18 and older (especially pharmacists) have to know how to distinguish between inaccurate and reliable information. Whether the sources were reliable or not, our results showed that 93.4% of interviewed pharmacists had good knowledge about COVID-19.

5. The authors did not elaborate in the results or discussion why there was a question in the survey about the pharmaceutical products usually purchased in drugstores.

We added some information in lines 189 - 191, 269 - 274. Vitamins and dietary supplements can be used for boosting the immune system. Antiseptics and hand sanitizers can be used for COVID-19 prevention as per the recommendations of the Vietnamese Ministry of Health. 

In the period of time of COVID-19 outbreak, the fact that these products were usually purchased showed that the COVID-19 prevention knowledge of the public was fairly good and activities involving propagandizing COVID-19 prevention of the government may be effective.

6. Table 4, C14: the reviewer is not sure of the meaning (or legality) when the authors mention asking patients about “medical declaration” when they purchase cold and flu medicine.

We revised this mistake. This is a translation mistake when we wrote the first draft. The phrase “medical declaration” was superseded by “taking notes”. For this question, we only wanted to ask about the attitude of pharmacists (not the practices). Several symptoms the same as COVID-19 contamination (such as cough and sore throat) can be omitted by patients if these symptoms are mild. Taking notes about the information (address, phone number) of customers purchasing medicines treating cold and flu can assist health agencies to trace and contact people with suspected COVID-19. In fact, at present, in Hanoi, the Department of Health required all pharmacists to collect information on these customers to contact them if necessary. In addition, everyone can access and easily do their medical declarations on the website of the government when necessary (only about 3 minutes). Citizens are recommended to do their medical declarations if suspecting of being infected with the COVID-19 virus.

7. Line 150, 151 mentions statistics about Iran with two different percentages.

Line 155: What do authors mean by “possessing many nimble and proficient authorities”?

We revised this mistake.

8. Line 170-171: The authors did not offer adequate explanation for their results on why assistant pharmacists with 3 years of training scored higher in COVID-19 knowledge than “Uni” with 5 years training, mentioning that “it seems that they (Unis) don’t care much about issues in the community in comparison with Col pharmacists”!

The explanation for our results on education level can be seen in lines 222 - 239.

9. The reviewer thinks the statement in line 220 on how paper surveys “approach various kinds of respondents who are not technology savvy (like old people)” is considered ageism and should be modified or removed.

We revised this mistake and removed the example “old people”.

---

## [Decision Letter · Decision Letter 1]

29 Jun 2021

PONE-D-21-08000R1

Knowledge, attitude and practices of community pharmacists regarding COVID-19: a paper-based survey in Vietnam

PLOS ONE

Dear Dr. Dinh,

Thank you for submitting your manuscript to PLOS ONE. After careful consideration, we feel that it has merit but does not fully meet PLOS ONE’s publication criteria as it currently stands. Therefore, we invite you to submit a revised version of the manuscript that addresses the points raised during the review process.

We look forward to receiving your revised manuscript.

Kind regards,

Bing Xue, Ph.D.

Academic Editor

PLOS ONE

Journal Requirements:

Reviewers' comments:

Reviewer's Responses to Questions

**Comments to the Author**

1. If the authors have adequately addressed your comments raised in a previous round of review and you feel that this manuscript is now acceptable for publication, you may indicate that here to bypass the “Comments to the Author” section, enter your conflict of interest statement in the “Confidential to Editor” section, and submit your "Accept" recommendation.

Reviewer #1: All comments have been addressed

Reviewer #2: (No Response)

2. Is the manuscript technically sound, and do the data support the conclusions?

Reviewer #1: Yes

Reviewer #2: Partly

3. Has the statistical analysis been performed appropriately and rigorously? 

Reviewer #1: I Don't Know

Reviewer #2: I Don't Know

4. Have the authors made all data underlying the findings in their manuscript fully available?

Reviewer #1: Yes

Reviewer #2: Yes

5. Is the manuscript presented in an intelligible fashion and written in standard English?

Reviewer #1: Yes

Reviewer #2: Yes

6. Review Comments to the Author

Reviewer #1: (No Response)

Reviewer #2: Thank you for addressing most of the previous comments. I have two more comments due to the changes in the manuscript:

1. The statement "In addition, a majority of Uni pharmacists who directly sell medicines in medicine outlets are graduates from private universities of pharmacy" is not supported by any reference. This information is also not captured in the survey and cannot be verified. Accordingly the explanation in lines 232-237 should be removed.

2. Table 3 in the changes tracked version contain the name of the provinces which matches the data analysis. This got dropped in the cleaned up version of the paper! Please make sure the final version has the breakdown of the region by the provinces.

7. PLOS authors have the option to publish the peer review history of their article (what does this mean?). If published, this will include your full peer review and any attached files.

Reviewer #1: No

Reviewer #2: No

---

## [Author Response · Author response to Decision Letter 1]

2 Jul 2021

Journal Requirements

The first reference (from WHO) was removed and another article with similar information was cited.

Regarding the second, the third, and the 6th references, some new information involving COVID-19 confirmed cases, deaths (in Vietnam and global), and COVID-19 vaccines was updated in the revised manuscript.

We checked twice whether or not articles were retracted (in Pubmed and Google Scholar). If any reference should be removed, please let us know.

Review Comments to the Author

Reviewer #2:

1. The statement "In addition, a majority of Uni pharmacists who directly sell medicines in medicine outlets are graduates from private universities of pharmacy" is not supported by any reference. This information is also not captured in the survey and cannot be verified. Accordingly the explanation in lines 232-237 should be removed.

Although that is the reality in our country, we do not have any references to cite. Some information in lines 232-237 is quite sensitive. Therefore, we removed these lines. Thank reviewer #2 for this practical advice.

2. Table 3 in the changes tracked version contain the name of the provinces which matches the data analysis. This got dropped in the cleaned up version of the paper! Please make sure the final version has the breakdown of the region by the provinces.

We added the names of provinces for each region in Table 3. In Table 3, we did not analyze the COVID-19 knowledge scores of pharmacists in each province because they were presented in Figure 1 with box plots and lines 213-220.

---

## [Decision Letter · Decision Letter 2]

16 Jul 2021

Knowledge, attitude and practices of community pharmacists regarding COVID-19: a paper-based survey in Vietnam

PONE-D-21-08000R2

Dear Dr. Dinh,

We’re pleased to inform you that your manuscript has been judged scientifically suitable for publication and will be formally accepted for publication once it meets all outstanding technical requirements.

Kind regards,

Bing Xue, Ph.D.

Academic Editor

PLOS ONE

Additional Editor Comments (optional):

Reviewers' comments:

Reviewer's Responses to Questions

**Comments to the Author**

1. If the authors have adequately addressed your comments raised in a previous round of review and you feel that this manuscript is now acceptable for publication, you may indicate that here to bypass the “Comments to the Author” section, enter your conflict of interest statement in the “Confidential to Editor” section, and submit your "Accept" recommendation.

Reviewer #2: All comments have been addressed

2. Is the manuscript technically sound, and do the data support the conclusions?

Reviewer #2: Yes

3. Has the statistical analysis been performed appropriately and rigorously? 

Reviewer #2: I Don't Know

4. Have the authors made all data underlying the findings in their manuscript fully available?

Reviewer #2: Yes

5. Is the manuscript presented in an intelligible fashion and written in standard English?

Reviewer #2: Yes

6. Review Comments to the Author

Reviewer #2: Thank you for revising the manuscript and for addressing all comments, no further recommendations from this reviewer.

7. PLOS authors have the option to publish the peer review history of their article (what does this mean?). If published, this will include your full peer review and any attached files.

Reviewer #2: No

---

## [Editor Report · Acceptance letter]

21 Jul 2021

PONE-D-21-08000R2 

Knowledge, attitude and practices of community pharmacists regarding COVID-19: a paper-based survey in Vietnam 

Dear Dr. Dinh:

I'm pleased to inform you that your manuscript has been deemed suitable for publication in PLOS ONE. Congratulations! Your manuscript is now with our production department. 

Kind regards, 

on behalf of

Professor Bing Xue 

Academic Editor

PLOS ONE